# Fine-Grained Butterfly Recognition via Peer Learning Network with Distribution-Aware Penalty Mechanism

**DOI:** 10.3390/ani12202884

**Published:** 2022-10-21

**Authors:** Chudong Xu, Runji Cai, Yuhao Xie, Huiyi Cai, Min Wang, Yuefang Gao, Xiaoming Ma

**Affiliations:** 1College of Electronic Engineering, South China Agricultural University, Guangzhou 510642, China; 2College of Mathematics and Informatics, South China Agricultural University, Guangzhou 510642, China; 3College of Plant Protection, South China Agricultural University, Guangzhou 510642, China; 4School of Public Administration, South China Agricultural University, Guangzhou 510642, China

**Keywords:** fine-grained recognition, long-tailed distribution, convolutional neural network, automatic species recognition, peer learning network

## Abstract

**Simple Summary:**

Automatic species recognition, such as butterflies or other insects, plays a crucial role in intelligent agricultural production management and the study of species diver-sity. However, the quite diverse and subtle interspecific differences and the long-tailed distribution of sample data in fine-grained species recognition are insufficient to learn robust feature representation and alleviate the bias and variance problems of the long-tailed classifier on insect recognition. The objective of this study is to develop a peer learning network with a distribution-aware penalty mechanism proposed to learn discriminative feature representation and mitigate the bias and variance problems in the long-tailed distribution. The results of various contrast experiments on collecting the butterfly-914 dataset show that the proposed PLN-DPM has a higher Rank-1 ac-curacy rate (86.2% on the butterfly dataset and 73.51% on the IP102 dataset). Addi-tionally, we deployed the PLN-DPM model on the smartphone app for butterfly recognition in a real-life environment.

**Abstract:**

Automatic species recognition plays a key role in intelligent agricultural production management and the study of species diversity. However, fine-grained species recognition is a challenging task due to the quite diverse and subtle interclass differences among species and the long-tailed distribution of sample data. In this work, a peer learning network with a distribution-aware penalty mechanism is proposed to address these challenges. Specifically, the proposed method employs the two-stream ResNeSt-50 as the backbone to obtain the initial predicted results. Then, the samples, which are selected from the instances with the same predicted labels by knowledge exchange strategy, are utilized to update the model parameters via the distribution-aware penalty mechanism to mitigate the bias and variance problems in the long-tailed distribution. By performing such adaptive interactive learning, the proposed method can effectively achieve improved recognition accuracy for head classes in long-tailed data and alleviate the adverse effect of many head samples relative to a few samples of the tail classes. To evaluate the proposed method, we construct a large-scale butterfly dataset (named Butterfly-914) that contains approximately 72,152 images belonging to 914 species and at least 20 images for each category. Exhaustive experiments are conducted to validate the efficiency of the proposed method from several perspectives. Moreover, the superior Top-1 accuracy rate (86.2%) achieved on the butterfly dataset demonstrates that the proposed method can be widely used for agricultural species identification and insect monitoring.

## 1. Introduction

Insects are the most diverse animal group in nature, accounting for approximately 47% of the more than 1.8 million species that have been described [1]. They are of great value in maintaining the ecological balance of nature, biological control and agricultural production. Among numerous insects, butterflies are more responsive to environmental changes than birds and other insects and are recognized as highly sensitive indicator species of environmental changes. As far as we know, there are nearly 20,000 species of butterflies in the world and China is rich in butterfly resources, with more than 2000 species. Due to this wide variety and the similar appearances of different genera and species, manual identification is time-consuming and labor-consuming and is highly dependent on the classification experience and skills of experts. For this reason, it is particularly necessary to carry out fine-grained automatic insect identification. Automatic species recognition can facilitate the intelligent management of agricultural species identification and benefit various applications for environmental monitoring, biological diversity protection, and sustainable agriculture development [2,3,4].

Instance recognition [5,6,7] has received much attention, and species recognition as its subset is more challenging. First, species recognition is a fine-grained recognition task that aims to discriminate similar subcategories belonging to the same superclass, e.g., recognizing hundreds of butterflies or moths. Due to the subtle and local distinctions between similar subcategories and the marked within-species variation of the samples belonging to the same subcategory, species recognition is difficult to perform, even for insect experts. In addition, the distribution of species categories is long-tailed, and a few common classes (a.k.a. head classes) contain many samples and many rare classes (a.k.a. tail classes) have only a few samples. In the case of species recognition, long-tailed distributions manifest as a few species being represented by many images and many species being represented by few images, it is difficult to learn robust and discriminative feature representation among these imbalanced data. Moreover, the number of categories of current species datasets is relatively small compared to nature’s diversity; e.g., the crop pest dataset [8] contains 4500 samples of 40 species, the IP102 [9] dataset includes 75,222 images of 102 species, Butterfly-200 [10] possesses 25,279 images of 200 species, and Caltech-UCSD birds-200-2011 [5] contains 11,788 images of 200 categories. Due to these challenges, little work has been dedicated to fine-grained species recognition under a natural long-tailed distribution. In this study, we explore a peer learning network with an adaptive model penalty mechanism that can learn discriminative and robust feature representation but also alleviate the bias and variance problems of long-tailed classifier through a parameter updating strategy. Moreover, a large-scale butterfly dataset is built for performance evaluation and further research.

Learning discriminative and subtle feature representation is vital for fine-grained species recognition task [11,12,13]. Instead of simply extracting global features, studies have explored local and multiscale features to further enhance the image representation of similar categories [11,14,15]. For example, Huang et al. [14] leveraged extra bounding box and part supervision to build part-based representation for capturing the subtle visual differences among specific parts. Zheng et al. [15] designed a progressive-attention convolutional neural network (CNN) that located subtle yet discriminative parts at multiple scales. Similarly, Wang et al. [16] proposed a feature fusion network with a patch detector that detected discriminative local patches without any part annotations and built a hierarchical representation by fusing global and local features to identify giant pandas. In contrast, Du et al. [17] proposed a progressive multigranularity (PMG) training approach for fine-grained feature learning that adopted a random jigsaw patch generator to learn features at specific granularities and fused multigranular features with a progressive training strategy. However, the image splitting operations and four training stages of the PMG method result in an exponential expansion of the training time.

Moreover, researchers have attempted to introduce extra guidance to learn more meaningful and semantic-related parts or characteristics. Chen et al. [10] integrated semantically structured category hierarchy information into a deep neural network to facilitate butterfly recognition. Furthermore, He et al. [18] utilized four types of media (images, text, audio, and video) to learn better common representations without discriminative treatment for fine-grained bird recognition. Du et al. [17] employed a saliency-guided discriminative learning network to simultaneously learn combined coarse-grained and fine-grained discriminative features in a multitask learning manner. Despite achieving positive performance for fine-grained species recognition, such approaches focus on tasks with roughly uniform class label distributions, and it remains challenging to effectively model the long-tailed distribution of fine-grained species for recognition purposes due to the extreme class imbalance issue.

To further mitigate the data imbalance problem in long-tailed distribution, class re-balancing strategies (e.g., cost-sensitive re-weighting, re-sampling, and transfer learning from head classes to tail classes) have been adopted in prior works that jointly learned representation and classifier to adjust the network training process [19,20,21,22]. To improve the representation learning process for the original data distribution, Zhou et al. [21] introduced a unified bilateral branch network (BBN) model to perform representation learning and classifier learning simultaneously and adjusted the bilateral learning procedure with a cumulative learning strategy to make it gradually pay attention to the tail data. Similarly, Work [23] systematically analyzed the performance of the above re-balancing strategies through a decoupled representation learning and classification scheme. A recent study [24] attempted to reduce the model bias and the computational cost by separately using a distribution-aware diversity loss and a dynamic expert routing module for learning high-quality representation.

Regarding long-tailed species classification, Wu et al. [9] studied the problem of subordinated insect recognition by introducing a long-tailed insect pest dataset (IP102) with 102 categories. Work [25] further proposed a weakly supervised multiple-instance learning method that identified image patches with saliency map guidance to achieve increased pest recognition performance on the public IP102 and citrus pest datasets. Despite achieving positive results, these existing studies mainly focused on basic-level visual recognition tasks (e.g., airplanes, birds, flowers, and frogs) that limited the applications of the classification models to specific domains, such as fine-grained object recognition. Moreover, related works [9,25] usually conducted experiments and analyses on the IP102 dataset or other smaller insect benchmarks that are prone to inducing model degradation in large-scale fine-grained species recognition tasks due to large class variations and subtle interclass differences.

To address the above issues, a peer learning network with the distribution-aware penalty mechanism is proposed to mitigate the problems concerning class imbalance and subtle interclass differences in long-tailed fine-grained species recognition. The framework cooperates with two existing identical CNN models to classify the input samples by utilizing different model parameters and alleviates the bias and variance of classifiers through a parameter update strategy. To this end, two pre-trained ResNeSt-50 modules are employed as the backbone of the peer learning module in the proposed framework to obtain the preliminarily predicted results. Then, the models are updated with different parameter strategies. The samples with different predicted labels are used to update the model parameters directly, while the suitable candidate samples with the same prediction labels are selected via a knowledge exchange strategy to update the model parameters of the backbone by using the distribution-aware penalty mechanism. By performing such adaptive collaborative learning, the proposed approach can effectively utilize the parameter update strategy to alleviate the class imbalance problem, and the recognition accuracy of head classes is further promoted by the interactive learning process of the two-stream backbone. Extensive experimental results obtained on the public IP102 dataset and our constructed butterfly dataset demonstrate that the proposed method obtains comparable results to those of not only the existing fine-grained image recognition methods but also the representative long-tailed recognition approaches.

The main contributions of this study can be summarized into three parts. First, we introduce a simple but universal peer learning network model with a distribution-aware penalty mechanism to mitigate the class imbalance problem, effectively exploring the optimal candidates with a parameter updating strategy to avoid learning corrupted information. Second, we construct a large-scale butterfly dataset (Butterfly-914) that exhibits a fine-grained characteristic and a natural long-tailed distribution. It can be used to effectively evaluate fine-grained and long-tailed butterfly recognition approaches and can also facilitate other research on species identification and verification. Finally, extensive experiments are conducted on the butterfly dataset to demonstrate the effectiveness of the proposed method and analyze the contribution of each component.

The rest of the paper is organized as follows. Section 2 presents the collection process of the butterfly dataset. Then, we introduce the details of the proposed method in Section 3. The experiments and applications are presented in Section 4 and Section 5, respectively, and finally, the conclusion and future work ideas are presented in Section 6.

## 2. Materials

To evaluate the efficiency of the proposed method for insect recognition, a large-scale butterfly dataset named Butterfly-914, which focuses on the fine-grained recognition of butterflies, is constructed. To this end, the following stages are implemented: (1) raw butterfly image collection; and (2) preliminary data preprocessing, where the butterfly dataset with accurate species labels is constructed by detecting the candidate images with tight bounding boxes around butterflies.

### 2.1. The Collection of Raw Butterfly Data

The butterfly image data are collected from two scenarios, including natural images in a field environment and standard images with butterflies in the form of specimens. The natural images are collected by searching the scientific names of butterflies on search engines, such as Google, Flicker, Bing, and Baidu. The standard images mainly come from the samples of a butterfly classification expert at the College of Plant Protection, South China Agricultural University. Images collected under these two scenarios are widely used in practical applications and contain challenging viewpoints, similarities, and various appearances over a wide range of resolutions. By utilizing such a strategy, numerous candidate images are collected for each species, and each image is saved to the directory with the corresponding Latin name.

After completing the preliminary data screening and data cleaning processes, a total of 68,385 butterfly images are obtained as the raw butterfly data. The raw dataset contains 914 species and covers 5 families, 29 subfamilies, and 332 genera. Figure 1 shows samples from the dataset, which is rather challenging due to the wide range of viewpoints, occlusion conditions, illumination changes, and background complexities. Specifically, the top four rows contain low-contrast butterfly examples under various viewpoints, illumination changes, and ambiguous backgrounds, while the last two rows present high-contrast butterfly samples with clean backgrounds.

### 2.2. Data Preprocessing

Most of the natural images in raw butterfly data contain many irrelevant objects, such as leaves, flowers, and other ambiguous background items, that adversely affect the discriminative feature representation process during model learning. Moreover, images may contain multiple butterflies belonging to the same category, which also affects the recognition performance. Based on the above observations, the raw butterfly data are processed by using an object detection strategy. Specifically, as shown in Figure 2, the raw butterfly samples are first input into the Faster R-CNN detection model, and then the butterfly images with bounding boxes are detected. Next, manual verification is conducted to delete the outlier samples, and the final butterfly dataset is built for use in the follow-up experiment.

### 2.3. The Butterfly Dataset

The dataset contains 72,152 images belonging to 914 butterfly categories, with more than 20 samples of each subject; these samples are detected from the raw butterfly data using an object detection model. Each image is organized by scientific classification (e.g., family, subfamily, genus, and species). For each species, the images produced from different scenarios are considered the same type.

The dataset has the following several appealing properties. First, with 72,152 labeled images, the proposed dataset is larger than those used in previous similar studies [10,26,27]. Second, the data collected from the indoor/outdoor environments contain abundant diversities, e.g., dramatic appearances, different viewpoints, heavy occlusion, and background clutter. This leads to a more challenging recognition problem. Moreover, the dataset is carefully identified by professional researchers and students to ensure its high reliability. Specifically, the butterfly images are annotated by the coauthor Min Wang and his students. Min is an expert in the classification and verification of butterflies at the College of Plant Protection, South China Agricultural University. In addition, the dataset not only has a fine-grained hierarchical taxonomy but also exhibits a natural long-tailed distribution. As illustrated in Figure 3a, each row separately denotes a different species within the same genus. Due to the subtle interclass distinctions, they cannot be classified by their global shapes. On the other hand, despite the samples in in each row of Figure 3b belonging to the same species, they exhibit substantially different appearances. The large variances in the same subcategory (Figure 3b) and the small variances among different subcategories (Figure 3b) make it highly challenging to distinguish them from each other, even for experts.

Apart from the biological diversity, the long-tailed distribution of the dataset further increases the difficulty of training a robust discriminative classifier. Figure 4 demonstrates the imbalanced distribution of the proposed butterfly dataset at different levels. As shown in Figure 4, a few butterfly species contain many samples, while most species contain only a few samples. In this case, it is more challenging to discriminate them because the few tail samples can be easily overwhelmed by the multitude of head samples.

## 3. Methods

In this section, a detailed description of the proposed peer learning network model is introduced. Moreover, a distribution-aware penalty mechanism is incorporated into the peer learning model to alleviate the bias and variance of the classifier.

### 3.1. Pipeline Overview

We aim to address the task of fine-grained butterfly recognition with a long-tailed distribution. To this end, a peer learning network with a distribution-aware penalty mechanism is developed; the network can learn robust discriminative fine-grained feature representation and mitigate the bias and variance of long-tailed classification problems utilizing a parameter updating strategy. The pipeline of the proposed method is shown in Figure 5.

The framework consists of three main modules, namely, a data enhancement module, a peer learning network, and the distribution-aware penalty learning mechanism with a knowledge exchange strategy. Specifically, the data derived from the input image are first enhanced to obtain more simple features (e.g., the global and local edge information) for model training and then input into the peer learning network to obtain the preliminary prediction results (The “p” in Figure 5 is the prediction results). Once the classification labels are acquired, those samples are divided into two groups based on whether the prediction results of each backbone model in the peer learning network are the same. The samples with different prediction labels (Figure 5a–d) mean both backbones are not good at classifying them, which are used to update the model parameters directly. Furthermore, the samples with the same prediction (in Figure 5e–g) and the prediction labels are wrong means these samples are affected by the dataset distribution. Thus, the strategy sets them as the candidate samples for knowledge exchange to obtain suitable model penalty samples. Finally, those penalty samples are used to update the model parameters of the peer learning network via the distribution-aware penalty mechanism. In this way, the proposed method can avoid learning ambiguous information by using the optimal candidates, further improving the recognition performance.

### 3.2. Modules of the Proposed Method

#### 3.2.1. Data Enhancement

As shown in Figure 5, data enhancement is performed on each image before feeding the images into the backbone of the peer learning network to improve the training effectiveness of the model. Specifically, due to the different resolutions of the samples in the dataset, each image is first resized to 256 × 256 to ensure that the images used for training and testing are of the same size. Then, those images are cropped to 224 × 224, and 50% random flipping is performed for each image to increase the randomness of the data. In addition, to eliminate the adverse effects caused by singular sample data, a normalization parameter from ImageNet is utilized to normalize the training data with their mean and standard deviation.

#### 3.2.2. The Peer Learning Network

As shown in Figure 5, inspired by the research of Sun et al. [28], the peer learning network module of our proposed framework employs two identical ResNeSt-50s to separately extract feature representation from data-enhanced butterfly images and predict the classification label of each sample. To this end, the pre-trained parameters without the fully connected layer are first loaded with ResNeSt-50 to acquire a preliminary feature extraction capability, as shown in Figure 6. Then, different normal weight initialization strategies are performed over the fully connected layer to ensure that the two backbones have different classification performances before the training stage. Specifically, Xavier normal initialization [29] and the Kaiming normal initialization [30] are separately employed for the initialization of the fully connected layer in the two-stream ResNeSt-50. The former makes the variance of the activation values and the variance of the state gradient of the fully connected layer consistent during propagation, while the latter enhances the generalization performance of the model by setting a nonzero derivative for the activation function. The normal distribution of the former is N=(0,std). Standard deviation (std) is defined below:(1)$$std=gain\times \sqrt{\frac{6}{fan\text{-}in+fan\text{-}out}}$$

Here, the value of the gain is 1. fan-in and fan-out represent the input dimensionality and the output dimensionality of the fully connected layer, respectively. The latter utilizes the same normal distribution as the former, with different std representations:(2)$$std=\sqrt{\frac{2}{\left(1+a^{2}\right)+fan\text{-}out}}\;$$
where the value a is set to 1, and fan-out is the output dimensionality of the fully connected layer.

The different initialization processes of classifiers make the two-stream ResNeSt-50 backbone have different classification performances before training. After the training data are propagated in the peer learning network, the two-stream ResNeSt-50 outputs the prediction label of each training sample. Then, those samples are divided into groups according to the predicted results, and all samples with the same prediction labels are set as the candidate samples for knowledge exchange, which maps the candidate samples to the loss function of the other ResNeSt-50 to obtain advice. The map of the knowledge exchange strategy is shown below:(3)$$map_{KE}=\{\left(L_{h_{1}}(x_{1}\right),L_{h_{2}}(x_{1}))\;\ldots \;,\;\left(L_{h_{1}}(x_{i}\right),L_{h_{2}}(x_{i}))\}\;$$

Here, $x_{i}$ represents the $i^{th}$ instance of the training set. $L_{h_{i}}$ represents the loss, which is calculated using the 1st or 2nd ResNeSt-50 in the two-stream backbone network. The losses of the two-stream ResNeSt-50 are mapped through the same samples. After obtaining the mapping, we sort $L_{h_{2}}$ from small to large in $map_{KE}$ to obtain $L_{h_{1}}$ with sequential changes. In this way, we also obtain the new order of $L_{h_{2}}$.

#### 3.2.3. Distribution-Aware Penalty Mechanism

Based on the predicted labels of the above peer learning module, the training data are divided into two groups: samples with different predicted labels and samples with the same predicted labels. Because the two streams of the backbone in the peer learning network module have the same structure, the images with different predicted labels signify that the prediction results of the two-stream backbone are wrong with a high probability. To improve the image recognition performance of the model, these data are directly used to update the model parameters. The samples with the same predicted labels represent that both backbone networks provide correct or wrong prediction simultaneously. To alleviate the incorrect predictions problem, the two-stream backbone developed in approach [28] utilized the knowledge exchange strategy to have the two streams learn from each other and updated the model parameters using the cross-entropy loss function.

Despite achieving acceptable performance, such a method is influenced by the distribution of the given dataset. To further reduce the misjudgment of the items with the same predicted labels, a distribution-aware penalty mechanism that uses the suitable samples selected by the knowledge exchange strategy for model parameter updating is introduced to mitigate the disturbance of the dataset distribution. To this end, inspired by the Destruction and Construction Learning (DCL) model in work [31], we set the classes with over 100 samples in the many-shot class and compute a chosen rate CR(V) based on this many-shot class. Then, the chosen rate is used to select samples that are useful for training the two-stream backbone module. CR(V) can be formulated as follows:(4)$$CR\left(V\right)=\frac{V_{m}}{V}$$

Here, V is the number of images in the dataset, and $V_{m}$ is the number of many-shot classes. We use these samples to train the two streams of the backbone.

An abstract of the main flow of the proposed knowledge exchange strategy with the distribution-aware penalty mechanism is presented in the pseudocode of Algorithm 1. The pipeline of the proposed mechanism is provided below.

(1) For a set of samples in a minibatch obeying a distribution $\tilde{D^{b}}$, two sets of losses ($L_{h_{1}}$, $L_{h_{2}}$) are obtained through the two-stream ResNeSt-50 module, and then the loss set by the samples is mapped for knowledge exchange using Equation (3);

(2) After acquiring the mapping and separately sorting the $L_{h_{2}}$ and $L_{h1}$ values from small to large based on their loss values, we use the chosen rate computed by Equation (4) to choose the suitable samples on $h_{1}$ with advice from $h_{2}$. Similarly, suitable samples are chosen on $h_{2}$ via advice from $h_{1}$;

(3) After performing knowledge exchange, the update sets $I_{1}$ and $I_{2}$ can be obtained. Then, we accumulate the update sets to separately obtain $Loss_{h1}\;\mathrm{and}\;Loss_{h2}$ for model parameter updating.

**Algorithm 1** Knowledge exchange strategy with distribution-aware penalty mechanism**input:** loss Ltotal; batch size b; chosen rate CR(V); two initial predictors h1,  h2 ∈H;   Lh1=0; Lh2=0.
**Output:**

$\;Loss_{h1};\;Loss_{h2}$

1 **for** t=1, 2, …, B **do**2  draw minibatch $\left(x_{1},\;\;y_{1}\right),\;\ldots ,\;\;\left(x_{b},\;\;y_{b}\right)\;\sim \;\tilde{D^{b}}$
  /* Map the loss of instance x through Equation (3) */3 $Map\left(\left\{L_{h_{1}}\left(x_{1}\right),\;\ldots \;,\;\;L_{h_{1}}\left(x_{b}\right)\right\},\;\;\left\{L_{h_{2}}(x_{1}),\ldots ,L_{h_{2}}(x_{b})\right\}\right)$
  /* Obtain a new sequence through the other stream network */4  $Sort\left(\left\{L_{h_{1}}\left(x_{1}\right),\;\ldots \;,\;\;L_{h_{1}}\left(x_{b}\right)\right\}\right)$ by $L_{h_{2}}$ in Map5   $\;\;\;\;\;Sort\left(\left\{L_{h_{2}}\left(x_{1}\right),\;\ldots \;,\;\;L_{h_{2}}\left(x_{b}\right)\right\}\right)$ by $L_{h_{1}}$ in Map6  let $I_{1}=\left\{\left(x_{i}\right):h_{1}\left(x_{i}\right)=h_{2}\left(x_{i}\right)\;AND\;\left|L_{h_{2}}\left(x_{i}\right)\right|\leq \left|L_{h_{2}}\left(x_{CR\left(V\right)\cdot b}\right)\right|\;\right\}$
7  let $I_{2}=\left\{\left(x_{i}\right):h_{1}\left(x_{i}\right)=h_{2}\left(x_{i}\right)\;AND\;\left|L_{h_{1}}\left(x_{i}\right)\right|\leq \left|L_{h_{1}}\left(x_{CR\left(V\right)\cdot b}\right)\right|\;\right\}$
  /* Accumulate the respective losses using Equation (11) */8 $Loss_{h1}+=L_{total}\left(I_{1}\right)$
9 $Loss_{h2}+=L_{total}\left(I_{2}\right)$
10 **end**11 **Return**$\;Loss_{h1};\;Loss_{h2}$


Through the above mapping and selection processes, some samples with small loss values can be obtained from one ResNeSt-50, but these samples might have large loss values in the other ResNeSt-50 of the two-stream module. By performing the above distribution-aware penalty learning procedure during updating, the proposed method can not only adapt to the complexity of the head data in the long-tailed data distribution but also enhance the robustness of the tail data, which will jointly improve the butterfly recognition performance of the model.

### 3.3. The Proposed Model Training Process

During the model training process of the proposed framework, the cross-entropy loss and distribution-aware loss are utilized to optimize the network parameters. Specifically, the samples with different prediction labels are used to update the parameters with the cross-entropy loss to describe the distance between the two probability distributions. The cross-entropy loss is defined as follows:(5)$$L_{classify}=-\overset{C}{\underset{i}{\sum }}y_{i}^{\prime }\log\left(y_{i}\right)$$
where $y_{i}$ is the predicted result and $y_{i}^{\prime }$ is the ground-truth label of the corresponding instance. C is the number of classes in the training dataset. For the samples with the same predicted labels, the distribution-aware loss is designed to maximize the Kullback–Leibler (*KL*) divergence between the different classification probabilities yielded for a sample *x* in class *y* over a total of c classes to alleviate the incorrect prediction problem. The definition of the distribution-aware loss is as follows:(6)$$L_{D\text{-}aware}=-D_{KL}(p^{1}\left(x,y\right)||p^{2}\left(x,y\right))$$

Here, $D_{KL}()$ and $p^{i}\left(x,y\right)$ represent the *KL* divergence and classifier, respectively, which are defined as follows:(7)$$D_{KL}(p||q)=\overset{C}{\underset{k=1}{\sum }}p_{k}log\left(\frac{p_{k}}{q_{k}}\right)$$
(8)$$p^{i}\left(x,y\right)=softmax\left(\left[\frac{D{\left(f\left(x\right)\right)}_{1}}{T_{1}}\ldots \;\frac{D{\left(f\left(x\right)\right)}_{c}}{T_{c}}\right]\right)$$

Here, $D{\left(f\left(x\right)\right)}_{k}$ represents the output value of the $k^{th}$ class object, and $T_{i}$ is the temperature function of the corresponding class that comes from work [32]. For a class k with $n_{k}$ samples, the smaller the $n_{k}$ and $T_{k}$ are, the more sensitive the classification probability is to a change in the feature f(x). The representation of $T_{k}$ is illustrated below:(9)$$T_{k}=\alpha \left(\beta_{k}+1-\max\left(\beta \right)\right)$$

Here,
(10)$$\beta_{k}=\lambda \cdot \frac{n_{k}}{\frac{1}{c}\sum_{s=1}^{C}n_{s}}+\left(1-\lambda \right)$$

$T_{k}$ scales linearly with the class size, ensuring that $\beta_{k}=1\;\mathrm{and}\;T_{k}=\alpha$ for a balanced set. The value of a is determined to be 1.5 in the experiment. This simple adaptation allows us to imbue the classifier with sufficient complexity for the head data and sufficient robustness for the tail data. For the samples with the same predicted labels, the final total classification loss $L_{total}$ is defined as follows:(11)$$L_{total}=L_{classify}+\zeta \cdot L_{D\text{-}aware}$$
where $L_{classify}$ and $L_{D\text{-}aware}$ are the cross-entropy loss obtained using Equation (5) and the distribution-aware diversity loss obtained via Equation (6), respectively. ζ is a hyperparameter that is restricted to the range of [0.2,0.8].

### 3.4. Experimental Setup

The experiment is conducted using an apparatus with an Intel^®^ Xeon(R) E5-2687w CPU @ 3.10 GHz, 64 GB of memory, and an NVIDIA GTX 3090 graphics card in an Ubuntu 18.04LTS system. Moreover, the experimental environment framework is based on Python 3.7 and PyTorch 1.10. The source code (The access date: 1 September 2022) will be made available on https://github.com/carajosaj/PLN-DPM.git.

In the experiments, two identical ResNeSt-50s are used as the backbone of the proposed model, and the initialized parameters of the two-stream backbone come from ImageNet. For the network input, the size of each image is set as 224 × 224, and the model is trained using the stochastic gradient descent (SGD) optimizer with a batch size of 8, a weight decay of 10^−5^ and a momentum of 0.9. The initial learning rate is set as 2 × 10^−3^, and it is divided by 10 after 30 epochs.

During the validation process of the recognition test, the Top-1 accuracy achieved over all classes is used for evaluation, and its formula is as follows:(12)$$Accuracy=\frac{TP+TN}{TP+FN+FP+TN}$$

Here, TP is the number of true-positive samples. FP is the number of false-positive samples. FN is the number of false-negative samples, and TN is the number of true-negative samples.

Additionally, we use the F1-score to evaluate the performance between models further. The function computes by precision and recall, which formula are as follow:(13)$$Precision_{k}=\frac{TP_{k}}{TP_{k}+FP_{k}}$$
which $TP_{k}$ is the number of true-positive samples from class *k* and the $FP_{k}$ is the number of false-positive samples from class *k*. $TP_{k}$ and $FP_{k}$ are used to compute the precision of the *k*th class.
(14)$$Recall_{k}=\frac{TP_{k}}{TP_{k}+FN_{k}}$$

Here, $FN_{k}$ is the number of false-negative samples from class *k*. $TP_{k}$ and $FN_{k}$ are used to compute the recall of the *k*th class.
(15)$$F1\text{-}{\mathrm{score}}_{k}=\frac{2\cdot Precision_{k}\cdot Recall_{k}}{Precision_{k}+Recall_{k}}$$

Furthermore, $F1\text{-}{\mathrm{score}}_{k}$ is computed by $Precision_{k}$ and $Recall_{k}$.
(16)$$F1\text{-}\mathrm{score}={\left(\frac{1}{n}\overset{n}{\underset{k}{\sum }}F1\text{-}{\mathrm{score}}_{k}\right)}^{2}$$

The final F1-score is obtained by averaging the *k*th F1-score under each class.

Moreover, to measure the performance of different output combinations of the two-stream recognition part in the peer learning network, two final predicted measurements $R_{1}$ and $R_{2}$ are employed, and their formulas are defined as follows:(17)$$R_{1}=argmax\left(y^{L-1}\right)$$
(18)$$R_{2}=argmax\left(y^{L-1}+y^{L-2}\right)$$
where *y* represents the predicted value of a single network in the two-stream part for the samples of all classes. L−i represents the *y* that comes from the $i^{th}$ fully connected layer.

## 4. Experiments

In this section, we analyze the recognition performance of the proposed method via extensive experiments and the Accuracy and F1-score are an average across five times trials. First, we use the ratio of 0.6:0.1:0.3 to divide datasets into the training, validation, and test sets before training and ensure they do not include very similar data. The training set is used to update the model parameters, the validation set is used to verify whether the model has completed training, and the test set is used to test the performance of trained model. Second, the existing leading generic object recognition methods are employed to evaluate the robustness of the proposed model. Third, we compare the proposed method with the representative fine-grained recognition approaches to assess its feature representation capability in various butterfly recognition challenges. Moreover, to illustrate the amelioration of the proposed method on the many-shot and few-shot classes in the long-tailed distribution, an experiment is conducted between several long-tailed models and our method. In contrast to the above comparisons based on our Butterfly-914 dataset, we further compare our results with the above methods on the fine-grained and long-tailed IP102 dataset to make another comparison. Furthermore, some comparative experiments are conducted to discuss the contribution of each module in our method.

### 4.1. Comparisons with the Generic Recognition Methods

In this experiment, the proposed model derived from Equation (12) is named Ours, whereas that obtained with Equation (13) is named Ours (Combined). We compare our proposed method with 11 generic object recognition methods, ResNet50 [33], ResNeXt-50 [34], EfficientNet-B0 [35], ResNeSt-50 [36], and PLN [28], as selected from among the existing leading generic object recognition models. It is worth mentioning that PLN is based on the idea of a two-stream backbone network, which has a similar mechanism to that of our proposed method. Except for the PLN model, the above compared models are based on a single backbone network.

The Top-1 accuracy plots of the different methods are presented in Figure 7, and the detailed results can be seen in Table 1. In Table 1, the first column represents the different recognition models, and the second column represents the number of feature extractors. The third column is the input resolution setting chosen before model training to ensure that the same feature extraction region is used, and the last column contains the Top-1 precision results of different approaches.

As shown in Table 1, the proposed method achieves superior performance compared to that of the state-of-the-art generic object recognition methods. Specifically, the performance of Ours is 85.3%, representing 4.4%, 4.0%, 2.7%, and 2.5% increases over the results of ResNet-50, ResNeXt-50, EfficientNet-B0 and ResNeSt-50, respectively. The Top-1 accuracy performance of Ours (Combined) is 86.2%, representing a 3.4% increase over that of the optimal ResNeSt-50 model. The reason for these findings can be attributed to the fact that the two-stream network can learn complementary predictions to further improve its recognition performance.

Despite utilizing a similar mechanism, the proposed method outperforms the PLN model by 2.2% and 3.1%. The superiority of our proposed method should be attributed to the distribution-aware penalty mechanism, which selects a suitable loss for model parameter updating to reduce the number of misjudged items with the same predicted labels. Moreover, the backbone ResNeSt-50 model, which adopts the split-attention block and comparatively more convolution operations, further improves the robustness of discriminative features. In addition, Ours (Combined) has a higher accuracy rate, with a 0.9% higher Top-1 accuracy than that of Ours. This result can be attributed to the complementarity of the outputs of the two-stream network, which enhances the robustness of the model to ambiguous prediction labels.

### 4.2. Comparisons with the Fine-Grained Recognition Approaches

The goal of this experiment is to test and verify the performance of the proposed approach on the fine-grained recognition task. To this end, we compare the proposed method with two existing representative fine-grained methods on the Butterfly-914 dataset. One is the PMG model [17], which is based on partial attention for recognition, and the other is the HSE model [10], which simultaneously predicts the categories at all levels in the hierarchy and integrates the structured correlation information into a deep neural network.

The Top-1 accuracy vs. number of training epochs curves obtained on the testing set are presented in Figure 8. Compared with the above models, although the PMG model has fewer fluctuations due to the split operation, the proposed method can be trained more stably and converges faster.

Table 2 further presents a detailed report of the results of different methods. It can be found that the recognition performances of our proposed models are all superior to those of the fine-grained approaches. In terms of Top-1 accuracy, the performance of Ours is 85.3%, representing 2.2% and 1.5% increases over those of the PMG and HSE models, respectively. The performance of Ours (Combined) is 86.2%, which is a 3.1% improvement over that of PMG and a 2.4% improvement over that of the HSE model. The better performance of our proposed methods can be attributed to the fact that they can capture not only generic features but also discriminative fine-grained visual representations.

### 4.3. Comparisons with the Leading Long-Tailed Recognition Methods

Due to the characteristics of the long-tailed distribution of the proposed Butterfly-914 dataset, we compare our proposed methods (Ours and Ours (Combined)) with the existing state-of-the-art methods, such as BBN [21], DRC [23], and RIDE [24], based on their performance. As shown in Figure 9, the training curves of our proposed methods are smoother than those of other long-tailed recognition methods. Table 3 presents the feature extraction backbone and the recognition results of the above methods. As shown in Table 3, the proposed methods significantly outperform all these other methods.

Moreover, we further analyze how the proposed method achieves improved performance on long-tailed data by splitting the groups based on the samples of each class. Inspired by the DRC model, classes in the dataset with different numbers of butterfly samples are divided into three groups: many-shot (more than 100), medium-shot (30–100), and few-shot (20–30) groups. As shown in Table 4, the proposed method achieves the best performance on all three groups.

In the many-shot class, the performances of the proposed Ours and Ours (Combined) models are 89.75% and 90.17%, respectively, which are 2.48% and 2.9% superior to that of the DRC model, 1.73% and 2.15% better than that of the BBN method, and 1.02% and 1.44% superior to that of the RIDE model, respectively. In the medium-shot class, the performances of our proposed methods are 80.56% and 80.63%, respectively, which are 4.37% and 4.44% better than that of the DRC model, 3.46% and 3.53% superior to that of the BBN model, and 2.81% and 2.88% better than that of the RIDE model. For the few-shot class, the two performances are 75.57% and 76.53%, which are 6.54% and 7.5% better than that of the DRC model, 6.6% and 7.56% superior to that of the BBN model, and 5.77% and 6.73% better than that of the RIDE model.

In addition, it can be observed that our proposed method is within a 1.9% margin for the many-shot class. The reason for this might be that these models can learn discriminative feature representations when the sample sizes are large. The performance of the proposed method is significantly higher than that of the other long-tailed algorithms on the medium-shot and few-shot classes. This can be attributed to the two-stream backbone and the knowledge exchange strategy with the distribution-aware mechanism used for instance selection, which can effectively improve the performance of the model for medium-shot and few-shot classification, thus simultaneously avoiding the performance degradation of the many-shot class.

### 4.4. Comparisons with the Existing Methods on a Public Long-Tailed Dataset

The previous experiments have illustrated how the proposed method achieves improved recognition performance on our proposed Butterfly-914 dataset. For a fairer evaluation, the proposed method is tested and evaluated on the public IP102 dataset for insect pest recognition. IP102 is not only a fine-grained recognition dataset with a total of 75,222 images belonging to 102 categories; it also exhibits a natural long-tailed distribution. In this experiment, we evaluate the performance of the state-of-the-art fine-grained/long-tailed methods, including IP102 [9], PMG [17], BBN [21], DRC [23], RIDE [24], WSLG [25], and our proposed model, on the IP102 dataset. The PMG model is a leading fine-grained image classification approach. The DRC, BBN, and RIDE models are representative methods for long-tailed recognition, while WSLG, IP102, and our proposed method focus on object representation with fine-grained and long-tailed characteristics.

Table 5 shows the recognition performance of the above models. We can see that the proposed methods, Ours and Ours (Combined), separately achieve 72.63% and 73.51% recognition accuracies, which are significantly higher than the results of the other mainstream models. In contrast to the fine-grained PMG model, the proposed models improve the Top-1 recognition accuracy by at least 5.87% and 6.75%, respectively. Compared with the long-tailed methods, the proposed method (such as Ours) outperforms the DRC, BBN, and RIDE approaches by 20.42%, 4.84%, and 4.5%, respectively. Our methods also significantly outperform the WSLG and IP102 models for fine-grained classification with a long-tailed data distribution. These desirable results can be attributed to the more discriminative local feature learning of the two-stream backbone and the knowledge exchange strategy with the distribution-aware mechanism in our proposed framework.

### 4.5. Ablative Analysis of Each Module of the Proposed Model

As shown in the previous subsection, the two-stream backbone and distribution-aware penalty mechanism of the proposed model significantly improve its recognition performance. This subsection presents a further ablative analysis to evaluate the contribution of the components of proposed model by comparing the recognition performance achieved with and without the penalty mechanism and under different backbone combinations on the proposed Butterfly-914 dataset. The configurations and results of the tested models are presented in Table 6. “+” and “−” indicate that the corresponding model is trained with and without the distribution-aware penalty mechanism, respectively.

As shown in Table 6, with the same CNN backbone (ResNet-50), the recognition performance of the PLN model is 83.1%, which is 2.2% better than that of the ResNet-50 model in which a single backbone is used. The results demonstrate that the combination of multiple CNN backbones can help achieve better recognition performance than that yielded by utilizing a single backbone. Moreover, with the same two-stream ResNet-50, the Top-1 accuracy of Ours (PLN + DPM) is 84.8%, which is 1.7% better than that of the PLN model. Note that the backbone design in the model is similar to that of the PLN method. However, our model makes an innovative change via the knowledge exchange strategy based on the distribution-aware penalty mechanism, which outperforms the PLN framework by a critical margin. This result can be attributed to the fact that the distribution-aware penalty mechanism in the knowledge exchange strategy can reduce the number of misjudgments concerning items with the same predicted labels. In addition, our proposed model and the Ours (Combined) model both outperform the other models, which means that the improved backbone and the combination prediction of the two-stream classifier further facilitate the recognition performance.

## 5. A Smartphone App for Butterfly Recognition

In this section, a smartphone app is developed based on the proposed method to recognize butterflies in a real-life environment, and a mini WeChat program is introduced in detail. Figure 10 displays the entire process of the butterfly recognition application, which is device-agnostic and works on any phone.

Users input a butterfly image saved in their smartphone or take a photo into the app to obtain the hierarchical names of the included butterflies, such as their family, subfamily, genus, and species classifications. The graphical display is shown in Figure 11. The software implementation of the app involves butterfly detection, a metadata base, butterfly recognition (Figure 11a), scan the QR code on the left to access the app), and a cloud-based data storage server (Figure 11b). The app uses the proposed model with a recognition rate of 86.2%, and its processing time on a mobile phone is 0.932 s. At present, the app is used in many universities to help experts and butterfly enthusiasts identify species.

## 6. Conclusions

This paper proposes a novel peer learning network that cooperates with a distribution-aware penalty mechanism to alleviate the problems of fine-grained representation learning and long-tailed data distribution in fine-grained image recognition tasks. Based on the predicted class labels of all samples provided by the two-stream backbone learning process, the proposed model utilizes the knowledge exchange strategy with the distribution-aware penalty mechanism to select suitable penalty samples for mitigating the misjudgments concerning items with the same predicted labels. To evaluate the proposed approach, a large-scale fine-grained butterfly dataset with a long-tailed distribution, Butterfly-914, is constructed. Extensive experiments and thorough analyses conducted on the Butterfly-914 dataset and the public IP102 pest dataset demonstrate the superiority of the proposed model over the state-of-the-art fine-grained methods, long-tailed methods, and other existing object recognition competitors.

Moreover, a device-agnostic application is developed to recognize butterflies readily in real environments. This application will provide high accuracy, easy operation, simplicity, and a low-cost means for performing butterfly management in agriculture monitoring. Furthermore, an attention mechanism can be introduced in follow-up research to extract more subtle and rich feature representations to improve the accuracy achieved on the few-shot class. In addition, we will investigate how much improvement is achieved by incorporating the re-identification method.

## Figures and Tables

**Figure 1 animals-12-02884-f001:**
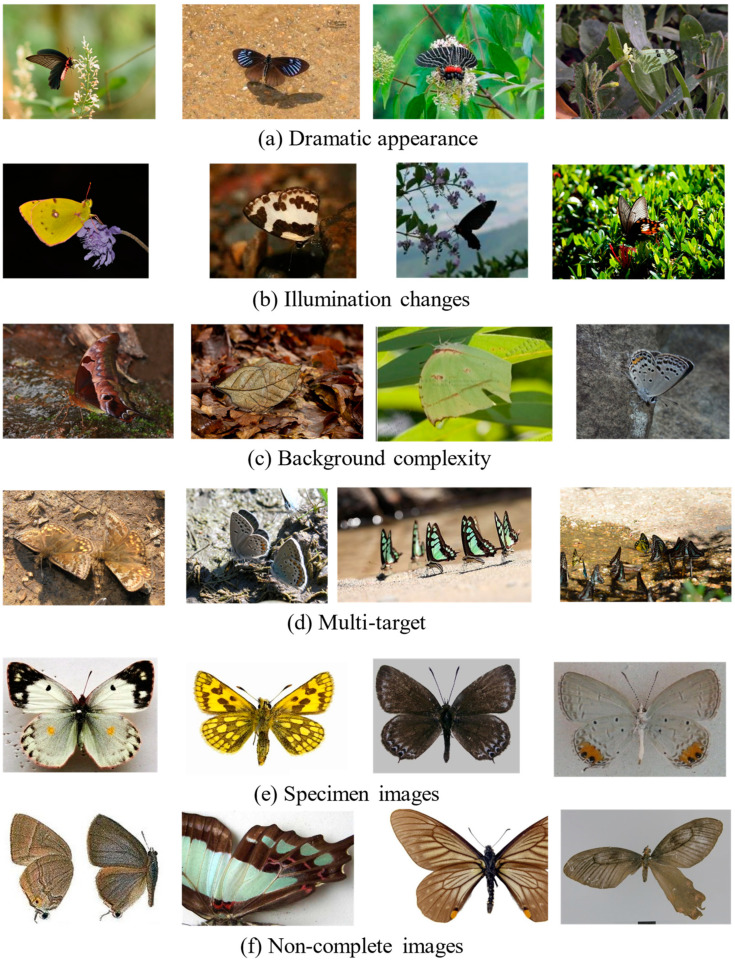
Examples of the raw butterfly data, which are challenging due to dramatic viewpoints, occlusion, and background complexity.

**Figure 2 animals-12-02884-f002:**
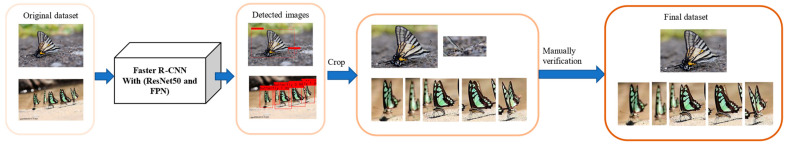
Generation process of the butterfly dataset.

**Figure 3 animals-12-02884-f003:**
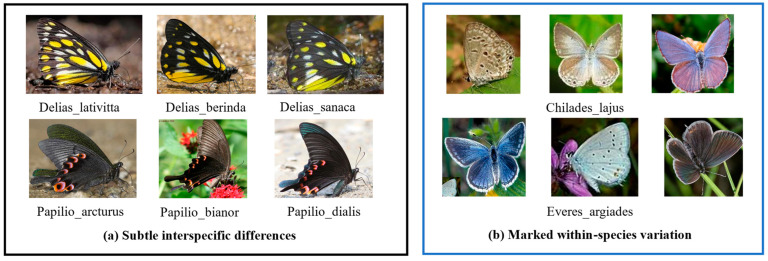
Illustrations of the subtle differences and substantially variances exhibited by butterflies.

**Figure 4 animals-12-02884-f004:**
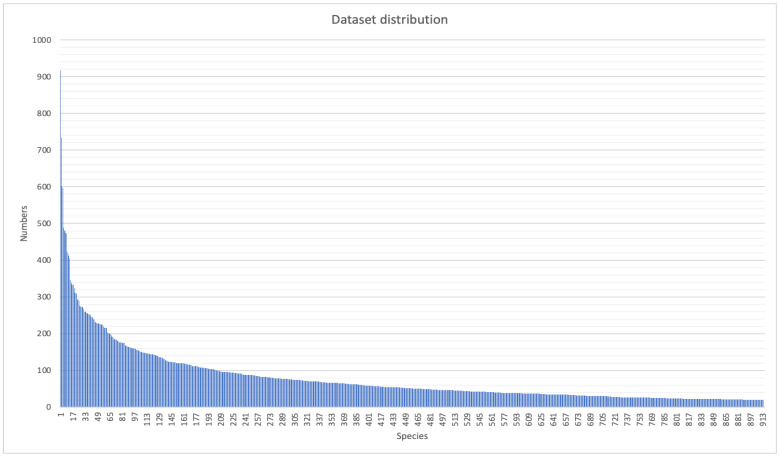
The sample number distribution of the proposed dataset.

**Figure 5 animals-12-02884-f005:**
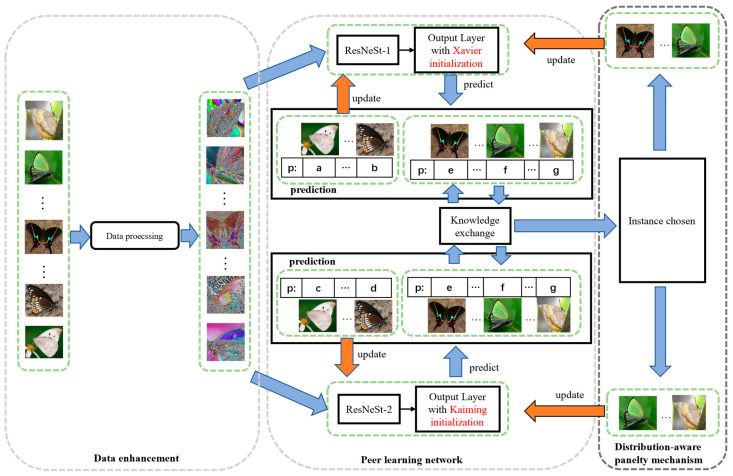
The architecture of the proposed method.

**Figure 6 animals-12-02884-f006:**
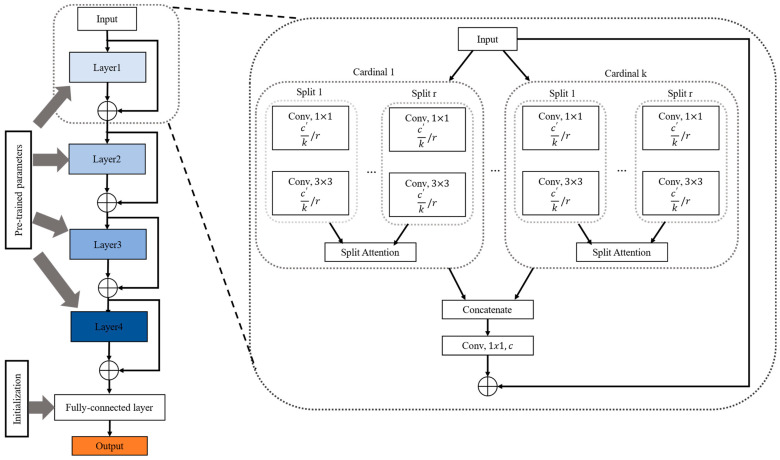
The structure and initialization of ResNeSt-50.

**Figure 7 animals-12-02884-f007:**
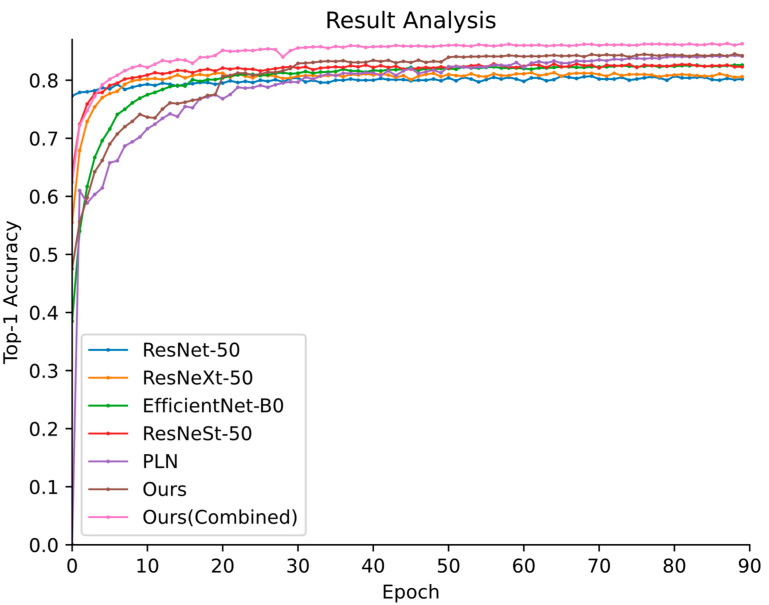
The Top-1 accuracy vs. number of training epochs curves yielded by different generic recognition methods on the Butterfly-914 testing set.

**Figure 8 animals-12-02884-f008:**
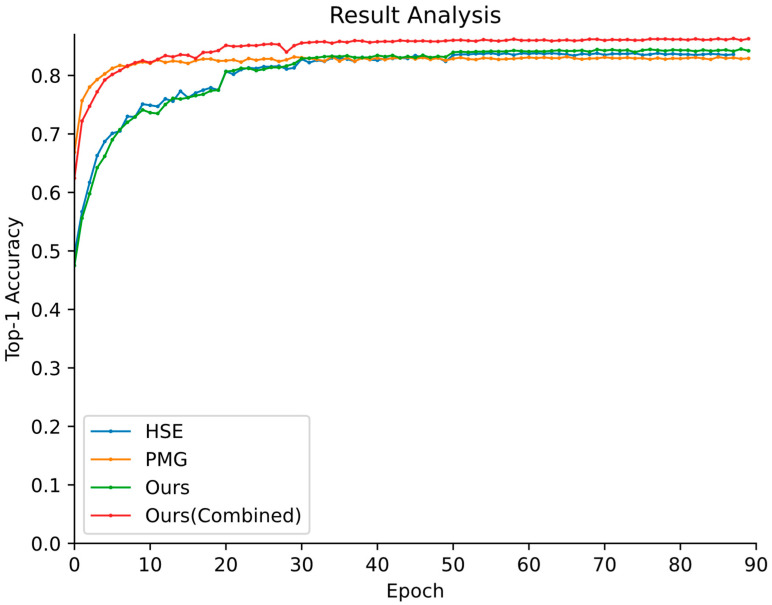
Top-1 accuracy vs. number of training epochs curves produced by different fine-grained models on the testing set.

**Figure 9 animals-12-02884-f009:**
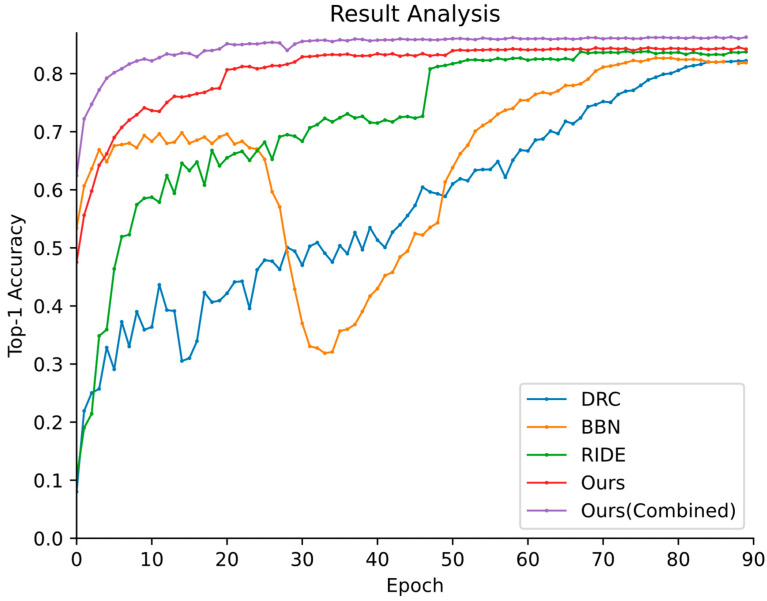
The Top-1 accuracy vs. number of training epochs curves produced by different long-tailed models on the testing set.

**Figure 10 animals-12-02884-f010:**
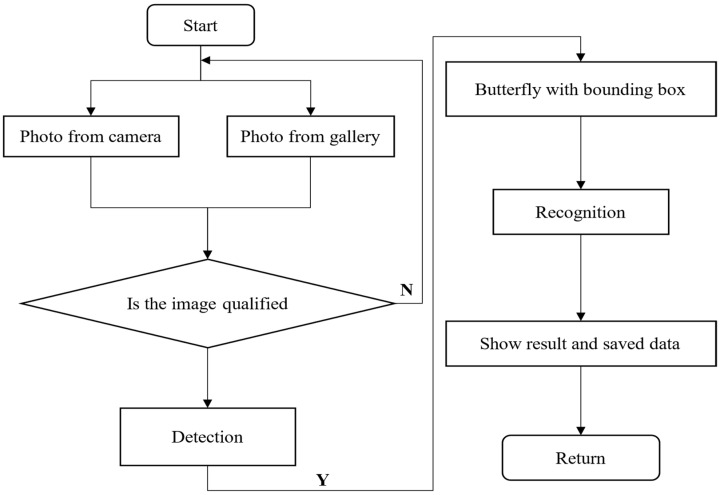
The entire process of application implementation.

**Figure 11 animals-12-02884-f011:**
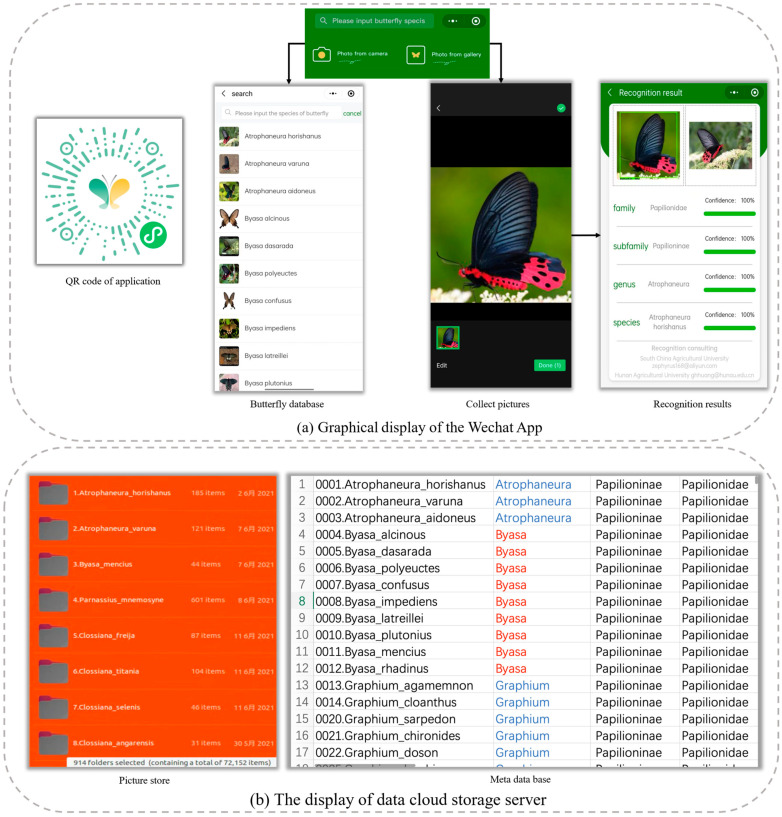
Schematic diagram of the butterfly recognition platform.

**Table 1 animals-12-02884-t001:** Top-1 accuracy and F1-score comparisons with existing leading generic object recognition models on the Butterfly-914 dataset. The highest score is shown in bold.

Methods	Backbone	F1-Score	Top-1 Accuracy (%)
ResNet-50	-	0.732	80.9
ResNeXt-50	-	0.744	81.3
EfficientNet-B0	-	0.756	82.6
ResNeSt-50	-	0.753	82.8
PLN	ResNet-50 × 2	0.764	83.1
Ours	ResNeSt-50 × 2	0.81	**85.3**
Ours (Combined)	ResNeSt-50 × 2	0.816	**86.2**

**Table 2 animals-12-02884-t002:** Top-1 accuracy and F1-score comparison with the fine-grained recognition methods. The highest score is shown in bold.

Model	Backbone	F1-Score	Top-1 Accuracy (%)
PMG (resize in 550 × 550)	Se-ResNeXt-50	0.769	83.1
HSE	ResNet-50	0.777	83.8
Ours	ResNeSt-50 × 2	0.81	**85.3**
Ours (Combined)	ResNeSt-50 × 2	0.816	**86.2**

**Table 3 animals-12-02884-t003:** Top-1 accuracy and F1-score comparisons with the long-tailed recognition methods. The highest score is shown in bold.

Model	Backbone	F1-Score	Top-1 Accuracy (%)
DRC	ResNeXt-50	0.735	82.2
BBN	ResNet-50	0.742	82.7
RIDE	ResNeXt-50	0.774	83.8
Ours	ResNeSt-50 × 2	0.81	**85.3**
Ours (Combined)	ResNeSt-50 × 2	0.816	**86.2**

**Table 4 animals-12-02884-t004:** Top-1 accuracy comparisons with the long-tailed methods on the many-shot, medium-shot, and few-shot classes. The highest score is shown in bold.

Model	Many-Shot	Medium-Shot	Few-Shot
DRC	87.27%	76.19%	69.03%
BBN	88.02%	77.1%	68.97%
RIDE	88.73%	77.75%	69.8%
Ours	**89.75%**	**80.56%**	**75.57%**
Ours (Combined)	**90.17%**	**80.63%**	**76.53%**

**Table 5 animals-12-02884-t005:** Top-1 accuracy comparisons with different fine-grained/long-tailed methods on the IP102 dataset. The highest score is shown in bold.

Model	Backbone	F1-Score	Top-1 Accuracy (%)
PMG	Se-ResNeXt-50	0.531	66.76
DRC	ResNeXt-50	0.383	52.21
BBN	ResNet-50	0.547	67.79
RIDE	ResNeXt-50	0.567	68.13
WSLG	Inception-v4	-	48.2
IP102	ResNet-50	-	49.4
Ours	ResNeSt-50 × 2	0.673	**72.63**
Ours (Combined)	ResNeSt-50 × 2	0.691	**73.51**

**Table 6 animals-12-02884-t006:** Ablative analysis conducted with/without different module combinations on the Butterfly-914 dataset. The highest score is shown in bold.

Model	DPM	Backbone	Top-1 Accuracy (%)
ResNet-50	−	−	80.9
PLN	−	ResNet-50 × 2	83.1
Ours (PLN + DPM)	+	ResNet-50 × 2	84.8
Ours	+	ResNeSt-50 × 2	**85.3**
Ours (Combined)	+	ResNeSt-50 × 2	**86.2**

## Data Availability

The butterfly and IP102 datasets presented in this study can be found on this link: [https://github.com/carajosaj/PLN-DPM.git] (accessed on 1 September 2022).

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
