# Peer review of "Fine-Grained Butterfly Recognition via Peer Learning Network with Distribution-Aware Penalty Mechanism"

_animals, 2022, doi:10.3390/ani12202884_

Round 1
Reviewer 1 Report
Chudong and colleagues present a novel method for identifying images of biological specimens to the species level. The work builds on previously described algorithms for pattern recognition and classification. Automated species identification, as described in the current work, is important because (1) biodiversity loss is a rising threat to ecosystem function and (2) the amount of digital data (i.e. photographs) is growing too large for manual (i.e. human) identification. The work is sound and would be a valuable contribution to the area of automatic species classification. There are some minor opportunities for clarification that I recommend be addressed before publication:
Line 12 (and elsewhere): The problem presented here is a classification problem, but the authors might consider avoiding using the term “interclass”, as the word “class” also has taxonomic implications. Since the taxonomic level of classification is at the species level, I think you could probably replace “subtle interclass differences among species” with “subtle differences among species” or even “subtle interspecific differences”, although the second option might be difficult for non-taxonomists. Other uses of the word “class” (e.g. “superclass”, line 47) should also be investigated and when possible, replaced.
Lines 34-35: A reference for the statement is warranted: “Among numerous insects, butterflies are more responsive to environmental changes than birds and other insects”
Line 35-36: A reference for the statement is warranted: “[butterflies] are recognized as highly sensitive indicator species of environmental changes.”
Line 39: I think you can just say “labor-consuming” (i.e. omit “time-consuming”).
Line 51: Some readers may benefit from a more explicit description of how these data are “long-tailed”. On line 53, after “only a few samples.” the authors could add: “In the case of species recognition, long-tailed distributions manifest as a few species being represented by many images and many species being represented by few images.”
Line 224: Provide a reference to the URL where the butterfly dataset (Butterfly-914) can be accessed (either here or in the Data Availability section).
Figure 1: Replace caption for (c) with “Background complexity”
Figure 3: The figure legend should be updated for clarity. Try: Illustrations of butterflies highlighting (a) subtle differences among species and (b) dramatic within-species variation.
Line 224: Instead of “tail instances” and “head instances”, try “low-frequency instances” and “high-frequency instances”.
Figure 5: Could the figure be updated to indicate the two different initialization strategies in the peer learning network module? I think this is what the figure is trying to communicate by having the parallel images.
Figure 5: Indicate that 256x256 refers to pixels (e.g. “resized to 256x256px”)
Lines 263-297. Section 3.2.2 describes the peer learning network module and could be enhanced by adding more references to Figure 5. For example, I think the difference between images (a, b) and (e, f, g) are that (e, f, g) ended up with the same prediction labels (so they are set as candidate instances for knowledge exchange). To help your readers make this connection, referencing the figure in the text (and adding additional labeling if necessary) would be useful. For example, line 290 could be updated to “and all instances with the same prediction labels (Figure 5, e, f, g)” and line 300 to “two groups: samples with different predicted labels (Figure 5 a, b) and samples with the same predicted labels (Figure 5, e, f, g).”
Line 316 (and elsewhere): Replace “many-shot” with “high frequency”.
Line 375: This GitHub repository should be made public.
Lines 404-405: Revise closing sentence with “Furthermore, we assessed contributions of each model in our method with a series of ablative analyses.”
Line 407 (and elsewhere): Consider using a more descriptive term for the model presented here than “Ours”, such as PLN-DPM (Peer Learning Network with Distribution-aware Penalty Mechanism). I note this is the name of the GitHub repository.
Line 428: If these algorithms were all trained on the same machine, are there any data on compute times? It would be useful to note the cost in terms of compute time (if any) for higher accuracy.
Line 463: Two scores are shown in bold in Table 2, but they are not the same (85.3 < 86.2).
Line 484: I do not think any of the scores are shown in bold in Table 3.
Lines 491-500: This paragraph is a verbal description of Table 4 and could be omitted.
Line 484: I do not think any of the scores are shown in bold in Table 4.
Line 503: I don’t understand what the 1.9% margin refers to in the following statement: “our proposed method is within a 1.9% margin for the many-shot class.”
Line 506 (and elsewhere): The term “significantly” will imply statistical significance to many audiences, but I do not think there were any assessments of statistical difference among the algorithms. Consider removing “significantly” or using a different word.
Line 514: I do not think the evaluation on the butterfly data was unfair (implied by “a fairer evaluation”), so I recommend replacing “For a fairer evaluation” with “For another evaluation”.
Figure 10: The “Y” should label the edge between the diamond “Is the image qualified” and the rectangle “Detection”.
Figure 10: Could the authors indicate (with a dotted polygon?) which steps are part of the PLN-DPM algorithm?
Figure 11: I do not think Figure 11 b (display of data on cloud server) is necessary.
In summary, I enjoyed reading this work and think it will be a valuable contribution to the field of automated species identification. Following some minor revisions to improve clarity, I look forward to seeing it in print.
Sincerely,
Jeffrey C. Oliver
University of Arizona
Author Response
Response to Reviewer 1 Comments(Please see the revised manuscript in the attachment.)
Dear Jeffrey C. Oliver:
Thank you very much for the review of our manuscript. We have carefully reviewed all your comments and addressed the comments and suggestions in the revised manuscript. All modifications use the “Track Changes” so you can easily distinguish the revised information. Besides, revisions that use the “Track Changes” function to markup will makes line numbers not consecutive from page to page.
Point 1: Line 12 (and elsewhere): The problem presented here is a classification problem, but the authors might consider avoiding using the term “interclass”, as the word “class” also has taxonomic implications. Since the taxonomic level of classification is at the species level, I think you could probably replace “subtle interclass differences among species” with “subtle differences among species” or even “subtle interspecific differences”, although the second option might be difficult for non-taxonomists. Other uses of the word “class” (e.g. “superclass”, line 47) should also be investigated and when possible, replaced.
Response 1: Thank you for your suggestion. Line 12, Lines 170-171, and Line 174 have replaced the “subtle interclass differences among species” with “subtle interspecific differences”. moreover, Line 279 has replaced “subtle class distinctions” with “subtle interspecific differences”, and Line 54 has replaced “superclass” with “class”.
Point 2: Lines 34-35: A reference for the statement is warranted: “Among numerous insects, butterflies are more responsive to environmental changes than birds and other insects”
Line 35-36: A reference for the statement is warranted: “[butterflies] are recognized as highly sensitive indicator species of environmental changes.”
Response 2: Thank you for your suggestion. The reference titled “Comparative losses of British butterflies, birds, and plants and the global extinction crisis” has been added in Line 36 for the warranted statement.
Point 3: Line 39: I think you can just say “labor-consuming” (i.e. omit “time-consuming”).
Response 3: Thank you so much for your kind reminding. The “time-consuming and” in Line 39 has been removed.
Point 4: Line 51: Some readers may benefit from a more explicit description of how these data are “long-tailed”. On line 53, after “only a few samples.” the authors could add: “In the case of species recognition, long-tailed distributions manifest as a few species being represented by many images and many species being represented by few images.”
Response 4: Thank you for your suggestion. On Line 60, after “only a few samples.” “In the case of species recognition, long-tailed distributions manifest as a few species being represented by many images and many species being represented by few images.” has been added.
Point 5: Line 224: Provide a reference to the URL where the butterfly dataset (Butterfly-914) can be accessed (either here or in the Data Availability section).
Response 5: Thank you very much for your kind reminder. We will make our codes and part of the butterfly dataset public before the manuscript accept (we do not own the copyright of some data).
Point 6: Figure 1: Replace caption for (c) with “Background complexity”
Response 6: Thank you very much for your suggestion. The caption for Figure 1 (c) has been replaced with “Background complexity”.
Point 7: Figure 3: The figure legend should be updated for clarity. Try: Illustrations of butterflies highlighting (a) subtle differences among species and (b) dramatic within-species variation.
Response 7: Thank you very much for your suggestion. The caption for Figure 3 (a) has been replaced with “Subtle interspecific differences” and (b) has been replaced with “Dramatic within-species variation”.
Point 8: Line 224: Instead of “tail instances” and “head instances”, try “low-frequency instances” and “high-frequency instances”.
Response 8: Thank you for your suggestion. The ”tail instances” in manuscript have been replaced with “low-frequency instances”, The “head instances” have been replaced with “high-frequency instances”.
Point 9: Figure 5: Could the figure be updated to indicate the two different initialization strategies in the peer learning network module? I think this is what the figure is trying to communicate by having the parallel images.
Response 9: Thank you for your suggestion. The figure has been updated to indicate the two different initialization strategies on the Output Layer in the peer learning network module.
Point 10: Figure 5: Indicate that 256x256 refers to pixels (e.g. “resized to 256x256px”)
Response 10: Thank you very much for your kind reminder. The resolutions in Line 330, 331, and 462 have been refered to pixels by adding “px”.
Point 11: Lines 263-297. Section 3.2.2 describes the peer learning network module and could be enhanced by adding more references to Figure 5. For example, I think the difference between images (a, b) and (e, f, g) are that (e, f, g) ended up with the same prediction labels (so they are set as candidate instances for knowledge exchange). To help your readers make this connection, referencing the figure in the text (and adding additional labeling if necessary) would be useful. For example, line 290 could be updated to “and all instances with the same prediction labels (Figure 5, e, f, g)” and line 300 to “two groups: samples with different predicted labels (Figure 5 a, b) and samples with the same predicted labels (Figure 5, e, f, g).”
Response 11: Thank you for your suggestion. The Line 313 has been updated to “obtain the preliminary prediction results (The p in Fig 5 means prediction results)”.
The Line 316 has been updated to “different prediction labels (Fig 5 “a, b” and “c, d”)”.
The Line 317-318 has been updated to “the same prediction results (Fig 5 “e, f, g”)”.
Point 12: Line 316 (and elsewhere): Replace “many-shot” with “high frequency”.
Response 12: Thank you for your suggestion. The “many-shot” in manuscript have been replaced with “high frequency”. The “few-shot” have been replaced with “low frequency”. The “medium-shot” have been replaced with “medium frequency”
Point 13: Line 375: This GitHub repository should be made public.
Response 13: Thank you very much for your kind reminder. We will make our GitHub repository and part of the butterfly dataset public before the manuscript accept (we do not own the copyright of some data).
Point 14: Lines 404-405: Revise closing sentence with “Furthermore, we assessed contributions of each model in our method with a series of ablative analyses.”
Response 14: Thank you for your suggestion. The closing sentence has been revised with “Furthermore, we assessed contributions of each model in our method with a series of ablative analyses.”.
Point 15: Line 407 (and elsewhere): Consider using a more descriptive term for the model presented here than “Ours”, such as PLN-DPM (Peer Learning Network with Distribution-aware Penalty Mechanism). I note this is the name of the GitHub repository.
Response 15: Thank you for your suggestion. We have revised “Ours” with “PLN-DPM” in the manuscript.
Point 16: Line 428: If these algorithms were all trained on the same machine, are there any data on compute times? It would be useful to note the cost in terms of compute time (if any) for higher accuracy.
Response 16: The experimental configuration has been detailed in Lines 455-458 (3.4. The experimental setup), and the experimental equipment (such as different GPUs) with different training times are different. There we think it is unnecessary to add the compute time. In addition, we indicate the test time on the smartphone app in Line 765(0.932s), which is helpful for readers.
Point 17: Line 463: Two scores are shown in bold in Table 2, but they are not the same (85.3 < 86.2).
Response 17: Thank you very much for your kind reminder. We have been bolded the “86.2” that is shown in Table 2.
Point 18: Line 484: I do not think any of the scores are shown in bold in Table 3.
Response 18: Thank you very much for your kind reminder. We have been bolded the “86.2” that is shown in Table 3.
Point 19: Lines 491-500: This paragraph is a verbal description of Table 4 and could be omitted.
Response 19: Thank you for your suggestion. We have been deleted these contents in Line 491-500 in revised manuscript.
Point 20: Line 484: I do not think any of the scores are shown in bold in Table 4.
Response 20: Thank you very much for your kind reminder. We have been bolded the combined results that are shown in Table 4.
Point 21: Line 503: I don’t understand what the 1.9% margin refers to in the following statement: “our proposed method is within a 1.9% margin for the many-shot class.”
Response 21: Thank you for your suggestion. We have been revised with ”our proposed method also improves the recognition accuracy of high-frequency classes”.
Point 22: Line 506 (and elsewhere): The term “significantly” will imply statistical significance to many audiences, but I do not think there were any assessments of statistical difference among the algorithms. Consider removing “significantly” or using a different word.
Response 22: Thank you for your suggestion. We have been removed the term “significantly” in revised manuscript.
Point 23: Line 514: I do not think the evaluation on the butterfly data was unfair (implied by “a fairer evaluation”), so I recommend replacing “For a fairer evaluation” with “For another evaluation”.
Response 23: Thank you for your suggestion. The “For a fairer evaluation” in manuscript have been replaced with “For another evaluation”.
Point 24: Figure 10: The “Y” should label the edge between the diamond “Is the image qualified” and the rectangle “Detection”.
Figure 10: Could the authors indicate (with a dotted polygon?) which steps are part of the PLN-DPM algorithm?
Response 24: Thank you very much for your kind reminder. We have been corrected the “Y” position and indicated the part of the PLN-DPM algorithm in Figure 10.
Point 25: Figure 11: I do not think Figure 11 b (display of data on cloud server) is necessary.
Response 25: Thank you for your suggestion. We have been removed the Figure 11 (b) in revised manuscript.
Sincerely,
Chudong Xu, Runji Cai, Yuhao Xie, Huiyi Cai, Min Wang, Yuefang Gao, Xiaoming Ma

Reviewer 2 Report
Thank you for your work and detailed manuscript explaining your experiments. I will try to provide you with suggestions to correct or improve it based on my expertise on the subject. I hope that my critical comments are detailed and clear enough to help you. I denote the lines of your manuscript's pdf with the letter "L" followed by the corresponding number. In the end, I provide my more general comments on your work.
L31: groups
L31-32: Add reference
L37: "where with China is being rich in butterfly resources"
L40: "For this reason, it is particularly necessary..."
L43: such as for
L45: TERMINOLOGY: Text and References [4-6] are vague for describing the already broad task of "instance recognition" and comparing it with "species recognition". Either clearly define what you describe as "instance recognition" or perhaps avoid the comparison altogether. Species recognition can be considered a subset of "instance recognition" per your references and L49 where you refer to "instances"?
L53: What do you mean by "few tails samples can be easily affected by many head instances"? How are they "affected"?
L55-59: The sizes are not small compared to other datasets, but maybe you mean they are small compared to nature's diversity? Or maybe that the ratio between species and number of samples is not good enough?
L68-69: Please indicate somewhere that you're talking about "object" parts in e.g. natural images or be more specific if possible (e.g. images of birds). "Part"/"parts" by itself as a word is unclear.
L109-110: "achieving positive progress" is a bit of rare wording, maybe use "outcomes" or "results"
L157: Section 2.1. Please mention the number of images gathered with each of the two scenarios. In L172 you mention 68,385 images but in L193 you mention 172,152 so it's not very clear where the discrepancy came from.
GENERAL SUGGESTION: Is "dramatic" standard terminology here (L203)? It seems that you use it in different ways. If it's not standard terminology, try to use "pronounced" or "marked" instead e.g. in Figure 3 (b). In another example, in L213 you could use "significantly"/"noticeably"/"substantially" instead of dramatically which is a word loaded with emotion.
L210: with within
L220: learning training
L222: The word instance is correct, but you could also use "sample" to avoid confusion with your earlier "instance learning" use.
L225: Maybe Methods instead of Method
L238-240: What is the purpose of this enhancement? It is not clear why you don't feed the original images to ResNeSt-50. L253, section 3.2.1 doesn't explain your reasoning either.
L241: It is not clear from your figure 5 how the classification labels are acquired. Maybe add a human icon to pinpoint human involvement in the process. Try to make the rightmost part of that figure more concise.
L243: I would recommend to add your reasoning behind each step. Explain , for example, why the "instances with different prediction labels are used to directly update the model parameters".
L258: Does random flipping get applied offline or during training? Please specify and mention whether you use other augmentation techniques and your reasoning behind them.
L269: Don't you mean "weight" initialization strategies?
L277: Use the full notation for standard deviation and then use the abbreviation std. Also "The std is defined as follows"
L288: Now you use samples while earlier you used "instances" for the same thing. To improve readability please stay consistent with one term. I recommend "samples".
L294: "the ith ResNeSt-50" Don't you only use 2? Index "i" is defined as 1 to the number of training data according to L293. Say instead "using 1st or 2nd ResNeSt-50".
L306: predictions
L310: Say why is it influenced by the distribution of the given dataset.
L320: Sentence is not clear (to learn from each other maybe?)
L332: Is Lossh1different to Lh1? If not, please stay consistent and use Lh1
Algorithm 1: Please remove horizontal lines or try to reformat this table in a more readable way. The vertical indentation line characters are also confusing.
Section 4.1: Is there a reason to compare with so many models?
Figure 7: it is overloaded and it's hard to see. Also, is it validation/test or training accuracy? Please mention that explicitly. Moreover, I suggest that you select a smaller sample of models (like 3-4 different ones), change the dotted-line to simple line or use smaller dots, remove the grid lines or make them thinner and remove the top and right border of your axes (e.g. by using seaborn.despine() ). The y axis should also start from 0 (now it includes negative values).
Table1: The column Resize doesn't offer any added value since it's always the same. Only 2 other models from different tables differ in input size, so you could add their input size in parenthesis next to their name and/or specify this discrepancy in writing.
Section 4.1: How did you split into your various Training/Validation(/Test) sets? You only mention a "testing set". Is that used during training? (common notation is to call it a validation set in that case)
If you use your testing set during training it also implies that it is not an unbiased testing set since your model changes its parameters based on it. It would be best to mention this in your text.
Did you evaluate the models on the same Validation(/Test) sets to avoid randomly assigning easier/harder tasks to various models?
Ideally you should employ both a validation set during training and a test set for final evaluation. If you only employ a validation set, then make sure that you select it in an unbiased way and that it doesn't include data that is very similar to the training set; otherwise, if you split your data randomly you will get an overfitting performance competition.
In case you split your data randomly: Randomly splitting all your data leads to overoptimistic results but since it is common practice in AI it is generally accepted. However, you should mention your data splitting procedure explicitly.
Section 4.2: Same comments as 4.1.
Section 4.3: Same comments as 4.1.
Adjust Figures 8 and 9 according to the comments for Figure 7.
GENERAL COMMENTS:
- It is known that Neural networks are non-deterministic and training them multiple times gives different results in accuracy each time. How did you tackle this issue while comparing the differences in performance for your various models? Please state in your methods section whether the results you show are the outcome of a single training procedure for each model or an average (or maximum) across trials.
- The accuracy metric is known to favor the majority classes. I would recommend that you look into other performance metrics such as: balanced accuracy, Matthews correlation coefficient or f1-score. You can still show accuracies in the figures and add more proper metrics in the tables where you compare your models.
- How do you ensure that your model performs well across all classes (majority and minority ones)? A figure showing your models' performance across all classes (over and under 100 samples) would give a good insight on their performance. If that figure is difficult to make, you could instead select some majority and minority classes or produce a performance metric across them.
- Testing on more public datasets would bring more value and understanding regarding your models than comparing it with multiple other known model architectures.
Good luck!
Author Response
Response to Reviewer 2 Comments (Please see the revised manuscript in the attachment.)
Thank you very much for the review of our manuscript. We have carefully reviewed all your comments and addressed the comments and suggestions in the revised manuscript. All modifications use the “Track Changes” so you can easily distinguish the revised information. Besides, revisions that use the “Track Changes” function to markup will makes line numbers not consecutive from page to page.
Point 1: L31: groups
Response 1: Thank you for your reminding. The grammatical error has been revised.
Point 2: L31-32: Add reference
Response 2: Thank you so much for your reminding. The reference titled “The prevalence of single-specimen/locality species in insect taxonomy: an empirical analysis” has been added in Line 31-32 for the warranted statement.
Besides, the content in Line 31-32 have been revised with “Insects are the most diverse animal group in nature, accounting for approximately 47% of the more than 1.8 million species that have been described”.
Point 3: L37: "where with China is being rich in butterfly resources"
Response 3: Thank you for your reminding. The grammatical error has been revised.
Point 4: L40: "For this reason, it is particularly necessary..."
Response 4: Thank you so much for your suggestion. We have revised with “For this reason, it is particularly necessary...” in Line 40.
Point 5: L43: such as for
Response 5: Thank you for your reminding. The grammatical error has been revised.
Point 6: L45: TERMINOLOGY: Text and References [4-6] are vague for describing the already broad task of "instance recognition" and comparing it with "species recognition". Either clearly define what you describe as "instance recognition" or perhaps avoid the comparison altogether. Species recognition can be considered a subset of "instance recognition" per your references and L49 where you refer to "instances"?
Response 6: Thank you so much for your reminding. The content in Line 57-58 has been revised with “Instance recognition [5-7] has received much attention, and species recognition as its subset is more challenging”.
Point 7: L53: What do you mean by "few tails samples can be easily affected by many head instances"? How are they "affected"?
Response 7: To clarify the meaning of "few tails samples can be easily affected by many head instances," model learning continuously revises model weights by computing the loss between the predicted and actual label of each sample. Each image will affect the model weights during the training (learning) process. Therefore, many head instances can be more influential on the model weights.
Point 8: L55-59: The sizes are not small compared to other datasets, but maybe you mean they are small compared to nature's diversity? Or maybe that the ratio between species and number of samples is not good enough?
Response 8: Thank you so much for your reminding. We mean they are small compared to nature's diversity, and the content in Line 67-68 has revised with “the number of categories of current species datasets is relatively small compared to nature's diversity”
Point 9: L68-69: Please indicate somewhere that you're talking about "object" parts in e.g. natural images or be more specific if possible (e.g. images of birds). "Part"/"parts" by itself as a word is unclear.
Response 9: The term “part” means local feature,for example, butterfly wings, texture, morphology in the image are local features. “part-based” means model focus on local features for object recognition.
Point 10: L109-110: "achieving positive progress" is a bit of rare wording, maybe use "outcomes" or "results"
Response 10: Thank you so much for your suggestion. We have revised “progress” with “results”.
Point 11: L157: Section 2.1. Please mention the number of images gathered with each of the two scenarios. In L172 you mention 68,385 images but in L193 you mention 72,152 so it's not very clear where the discrepancy came from.
Response 11: Thank you for your suggestion. We are very sorry that we did not separate statistics on the number of images gathered with each of the two scenarios. Line 247-248 mentions that the 68,385 images from the data were collected from the two scenarios. Line 256-266 mentions that the 72,152 images came from deploying data processing on the 68,385 images.
Point 12: GENERAL SUGGESTION: Is "dramatic" standard terminology here (L203)? It seems that you use it in different ways. If it's not standard terminology, try to use "pronounced" or "marked" instead e.g. in Figure 3 (b). In another example, in L213 you could use "significantly"/"noticeably"/"substantially" instead of dramatically which is a word loaded with emotion.
Response 12: Thank you very much for your suggestion. The caption for Figure 3 (b) has been revised with “Marked within-species variation”.
The “dramatic variances” in Line 61 has been revised with “marked within-species variation”.
The “dramatically” in Line 295 has been revised with “substantially”.
Point 13: L210: with within
L220: learning training
L222: The word instance is correct, but you could also use "sample" to avoid confusion with your earlier "instance learning" use.
L225: Maybe Methods instead of Method
Response 13: Thank you for your reminding. The grammatical errors have been revised.
Point 14: L238-240: What is the purpose of this enhancement? It is not clear why you don't feed the original images to ResNeSt-50. L253, section 3.2.1 doesn't explain your reasoning either.
Response 14: The images fed into the network must be of the same resolution. Besides, the random flip allows the model to recognize pictures from different angles.
Point 15: L241: It is not clear from your figure 5 how the classification labels are acquired. Maybe add a human icon to pinpoint human involvement in the process. Try to make the rightmost part of that figure more concise.
Response 15: Thank you very much for your suggestion. We have updated the Figure 5 and the content in Line 325-338. Figure 5 shows the procession of training model, first, the image will obtain a predicted label by network, then the update strategy uses the predicted labels and the correct label to modify network weights.
Point 16: L243: I would recommend to add your reasoning behind each step. Explain , for example, why the "instances with different prediction labels are used to directly update the model parameters".
Response 16: Thank you very much for your suggestion. The samples with different prediction labels mean both backbones are not good at classifying them, directly updating model weights. Besides, the samples with the same prediction and the prediction labels are wrong means these samples are affected by the dataset distribution. We have updated the content in Lines 325-338.
Point 17: L258: Does random flipping get applied offline or during training? Please specify and mention whether you use other augmentation techniques and your reasoning behind them.
Response 17: The data enhancement gets applied before inputting it into the classification backbone. We did not use the other augmentation techniques. The reason of data enhancement has mentioned in Point 14.
Point 18: L269: Don't you mean "weight" initialization strategies?
Response 18: Yes, we mean the weight initialization strategies on full-connected layer.
Point 19: L277: Use the full notation for standard deviation and then use the abbreviation std. Also "The std is defined as follows"
Response 19: Thank you so much for your reminding. The full notation has been added.
Point 20: L288: Now you use samples while earlier you used "instances" for the same thing. To improve readability please stay consistent with one term. I recommend "samples".
Response 20: Thank you very much for your suggestion. The “instances” has been replaced with “samples” in the revised manuscript.
Point 21: L294: "the ith ResNeSt-50" Don't you only use 2? Index "i" is defined as 1 to the number of training data according to L293. Say instead "using 1st or 2nd ResNeSt-50".
Response 21: Thank you very much for your suggestion. The “ith” has been replaced with “1st and 2nd ” in the revised manuscript.
Point 22: L306: predictions
Response 22: Thank you for your reminding. The grammatical error has been revised.
Point 23: L310: Say why is it influenced by the distribution of the given dataset.
Response 23: The reason has mentioned at Point 7. Model learning is a process of continuously revising model parameters. Each image will affect the model parameters during the training (learning) process. Therefore, many head instances can be more influence on the model parameters. If the distribution is unbalance, the performance on tail samples will be affected by head samples.
Point 24: L320: Sentence is not clear (to learn from each other maybe?)
Response 24: Thank you very much for your suggestion. To make sentence clear, we have revised with “to train the two-stream backbone”.
Point 25: L332: Is Lossh1 different to Lh1? If not, please stay consistent and use Lh1
Response 25: Thank you for your reminding. The “Lossh1” has been revised.
Point 26: Algorithm 1: Please remove horizontal lines or try to reformat this table in a more readable way. The vertical indentation line characters are also confusing.
Response 26: Thank you very much for your suggestion. We have removed horizontal lines and vertical indentation line characters from Algorithm Table.
Point 27: Section 4.1: Is there a reason to compare with so many models?
Figure 7: it is overloaded and it's hard to see. Also, is it validation/test or training accuracy? Please mention that explicitly. Moreover, I suggest that you select a smaller sample of models (like 3-4 different ones), change the dotted-line to simple line or use smaller dots, remove the grid lines or make them thinner and remove the top and right border of your axes (e.g. by using seaborn.despine() ). The y axis should also start from 0 (now it includes negative values).
Response 27: Thank you very much for your suggestion. Lines 531-532 give the explanation that the models are the existing leading generic object recognition methods. L554 mention that results are the test accuracy. We have carefully considered your suggestion, and recvised Figure 7 and added F1-score in Table 1.
Point 28: Table1: The column Resize doesn't offer any added value since it's always the same. Only 2 other models from different tables differ in input size, so you could add their input size in parenthesis next to their name and/or specify this discrepancy in writing.
Response 28: Thank you very much for your suggestion. We have removed The column Resize and replaced with The column F1-score.
Point 29: Section 4.1: How did you split into your various Training/Validation(/Test) sets? You only mention a "testing set". Is that used during training? (common notation is to call it a validation set in that case)
If you use your testing set during training it also implies that it is not an unbiased testing set since your model changes its parameters based on it. It would be best to mention this in your text.
Did you evaluate the models on the same Validation(/Test) sets to avoid randomly assigning easier/harder tasks to various models?
Ideally you should employ both a validation set during training and a test set for final evaluation. If you only employ a validation set, then make sure that you select it in an unbiased way and that it doesn't include data that is very similar to the training set; otherwise, if you split your data randomly you will get an overfitting performance competition.
In case you split your data randomly: Randomly splitting all your data leads to overoptimistic results but since it is common practice in AI it is generally accepted. However, you should mention your data splitting procedure explicitly.
Response 29: Thank you very much for your suggestion. We have added the process about how to split in Lines 527-531.
Point 30: Section 4.2: Same comments as 4.1.
Section 4.3: Same comments as 4.1.
Adjust Figures 8 and 9 according to the comments for Figure 7.
Response 30: Thank you very much for your suggestion. We have carefully considered your suggestion that we revise Figures 8,9 and add the F1-score in Tables 2,3,5.
GENERAL COMMENTS:
Point 31: - It is known that Neural networks are non-deterministic and training them multiple times gives different results in accuracy each time. How did you tackle this issue while comparing the differences in performance for your various models? Please state in your methods section whether the results you show are the outcome of a single training procedure for each model or an average (or maximum) across trials.
Response 31: Thank you very much for your suggestion. We have added the statement of how to obtain the result in Line 527.
Point 32: - The accuracy metric is known to favor the majority classes. I would recommend that you look into other performance metrics such as: balanced accuracy, Matthews correlation coefficient or f1-score. You can still show accuracies in the figures and add more proper metrics in the tables where you compare your models.
Response 32: Thank you very much for your suggestion. We have carefully considered your suggestion that we revise Figures 7,8,9 and add the F1-score in Tables 1,2,3,5 in the revised manuscript.
Point 33: -How do you ensure that your model performs well across all classes (majority and minority ones)? A figure showing your models' performance across all classes (over and under 100 samples) would give a good insight on their performance. If that figure is difficult to make, you could instead select some majority and minority classes or produce a performance metric across them.
Response 33: Inspire by DRC [23], the average accuracy on the testing set allows us to ensure the model performs well across many-shot classes and few-shot classes.
Point 34: Testing on more public datasets would bring more value and understanding regarding your models than comparing it with multiple other known model architectures.
Response 34: Thank you for your kind suggestion. We have carefully studied the task you mentioned. In the manuscript, in addition to our dataset butterfly-914, we also deploy experiments on the public dataset (IP102) and we will continue to pay attention to this task and improve our model in the future.

Round 2
Reviewer 2 Report
Thank you for your nice work once again and for addressing most of my comments. Here are my further comments based on "author_response.docx"
Point1: I appreciate that you added the F1-score metric to give a better understanding on model performance, but you should also add its definition somewhere; perhaps after equation (12) where you define Accuracy.
Point2 (ex Point14): In my previous Point14 you mention only a resolution change, as a data enhancement strategy but what you show on Figure 5 seems like you perform some other transformation that affects the colors of the images. More specifically, on L248 you say "...the data derived from the input image are first enhanced to obtain more simple features..". What exactly do you do to "enhance" your input data? I can't find that anywhere on the paper. It looks like "Solarize" augmentation but I can't say for sure since the icons are small.
Point 3: L286 Please add the word "weight" --> "different normal weight initialization strategies" so that it's clearer for more readers.
Point 4: L292: shown as followbelow or "defined as follows" or "defined below"
Point 5 (ex-Point 7): L53: Like you said, model performance is the one that is affected. In this sentence you write "few tail samples can be easily affected by many head samples". Please clarify in the text that model performance is affected more by multiple head samples than it is by fewer tail samples.
Author Response
Point1: I appreciate that you added the F1-score metric to give a better understanding on model performance, but you should also add its definition somewhere; perhaps after equation (12) where you define Accuracy.
Response 1: Thank you so much for your reminding. The equation has been added after equation (12) in revised manuscript.
Point2 (ex Point14): In my previous Point14 you mention only a resolution change, as a data enhancement strategy but what you show on Figure 5 seems like you perform some other transformation that affects the colors of the images. More specifically, on L248 you say "...the data derived from the input image are first enhanced to obtain more simple features..". What exactly do you do to "enhance" your input data? I can't find that anywhere on the paper. It looks like "Solarize" augmentation but I can't say for sure since the icons are small.
Response 2: The images that be affected the colors in Figure 5 through the transformation in the final sentence “a normalization parameter from ImageNet is utilized to normalize the training data with their mean and standard deviation.” in section 3.2.1.
Obtaining more simple features means that the boundaries of the butterfly's wings and body are more obvious in the transformed image in Figure 5 after enhancement.
Point 3: L286 Please add the word "weight" --> "different normal weight initialization strategies" so that it's clearer for more readers.
Response 3: Thank you so much for your suggestion. We have been added “weight” in “different normal weight initialization strategies”.
Point 4: L292: shown as followbelow or "defined as follows" or "defined below"
Response 4: Thank you for your reminding. The grammatical errors have been revised.
Point 5 (ex-Point 7): L53: Like you said, model performance is the one that is affected. In this sentence you write "few tail samples can be easily affected by many head samples". Please clarify in the text that model performance is affected more by multiple head samples than it is by fewer tail samples.
Response 5: Thank you for your suggestion. For clarity, we have revised with “In the case of species recognition, long-tailed distributions manifest as a few species being represented by many images and many species being represented by few images.”.
